# Phase I/II trial of a peptide-based COVID-19 T-cell activator in patients with B-cell deficiency

T-cell immunity is central for control of COVID-19, particularly in patients incapable of mounting antibody responses. CoVac-1 is a peptide-based T-cell activator composed of SARS-CoV-2 epitopes with documented favorable safety profile and efficacy in terms of SARS-CoV-2-specific T-cell response. We here report a Phase I/II open-label trial (NCT04954469) in 54 patients with congenital or acquired B-cell deficiency receiving one subcutaneous CoVac-1 dose. Immunogenicity in terms of CoVac-1-induced T-cell responses and safety are the primary and secondary endpoints, respectively. No serious or grade 4 CoVac-1-related adverse events have been observed. Expected local granuloma formation has been observed in 94% of study subjects, whereas systemic reactogenicity has been mild or absent. SARS-CoV-2-specific T-cell responses have been induced in 86% of patients and are directed to multiple CoVac-1 peptides, not affected by any current Omicron variants and mediated by multifunctional T-helper 1 CD4[+] T cells. CoVac-1-induced T-cell responses have exceeded those directed to the spike protein after mRNA-based vaccination of B-cell deficient patients and immunocompetent COVID-19 convalescents with and without seroconversion. Overall, our data show that CoVac-1 induces broad and potent T-cell responses in patients with B-cell/antibody deficiency with a favorable safety profile, which warrants advancement to pivotal Phase III safety and efficacy evaluation. ClinicalTrials.gov identifier NCT04954469.

The coronavirus disease-19 (COVID-19) pandemic caused by severe acute respiratory syndrome coronavirus 2 (SARS-CoV-2) prompted the development of several vaccines which protect billions of people from severe course of disease in particular by induction of humoral, i.e., antibody-mediated immunity[1–4]. Patients unable to mount humoral immune responses, neither to natural infection nor to prophylactic vaccination, are at high risk for a dismal outcome of COVID-19[5–8]. This comprises individuals with congenital B-cell deficiency, but also cancer patients with disease or treatment related B-cell depletion. Beyond humoral immunity mediated by B cells, T cells are key for COVID-19 outcome and maintenance of immunity to SARS-CoV-2[9–16].

In a Phase I trial, our peptide-based T-cell activator CoVac-1 showed a favorable safety profile and induced broad and long-lasting T-cell

immunity that by far exceeded T-cell responses after SARS-CoV-2 infection as well as after vaccination with any approved vaccine[17]. CoVac-1 is a multi-peptide-based T-cell activator designed to induce, upon a single application, a broad and long-lasting SARS-CoV-2 T cell immunity resembling that acquired by natural infection[17]. It is composed of multiple SARS-CoV-2 human leukocyte antigen (HLA)-DR T-cell epitopes, which are derived from different viral proteins (spike, nucleocapsid, membrane, envelope, open reading frame (ORF) 8) that have been proven to be (i) frequently and HLA-independently recognized by T cells in convalescent individuals after COVID-19, (ii) of pathophysiological relevance for T-cell immunity to combat COVID-19, and (iii) to mediate long-term immunity after infection[9,10,18] and, thus, induce T-cell immunity that is independent of existing variants of concern (VOCs)[17].

✉ e-mail: juliane.walz@med.uni-tuebingen.de

We here report the results of the open-label Phase I/II trial evaluating immunogenicity along with safety and reactogenicity of CoVac-1 in the high-risk population of patients with congenital or acquired B-cell deficiency.

## Results

### Patients

From July 6th, 2021 to January 13th, 2022, a total of 94 patients with congenital or acquired B-cell deficiency underwent screening at three study sites in Germany. A total of 54 patients received CoVac-1, 14 patients in the Phase I safety run-in, and 40 patients in the subsequent Phase II part of the trial. 28% of patients were female. Median patient age was 61.8 (range 37–90) years. 93% of study patients suffered from cancer-related, acquired B-cell deficiency, with chronic lymphocytic leukemia (CLL, 30%), mantle cell lymphoma (MCL, 24%) and follicular lymphoma (FL, 20%) as the most common diagnoses. Application of an approved COVID-19 vaccine prior to study inclusion was reported for 83% of patients with a median of two vaccinations per patient (Supplementary Table 7). CD4+ T-cell counts in the study population ranged from 123 to 2501/μl (median 458/μl). All patients received one dose of CoVac-1 on day 1 and were available for safety analyses until day 56 (Fig. 1). 49 patients were eligible for immunogenicity analysis until day 28. One major protocol violation occurred (missed study visit day 28). Analyses of follow-up safety and long-term immunogenicity data (until month 6) are ongoing. Demographic and clinical characteristics of the patients are provided in Table 1 as well as in Supplementary Tables 7 and 8.

### Safety and reactogenicity

Data regarding solicited and unsolicited adverse events (AEs) were available for all patients from diary cards (for 28 days after CoVac-1 application) and safety visits (until day 56). No patient discontinued the trial because of an AE. No vaccine-related serious adverse events (SAEs) and no grade 4 AEs were reported. Until day 56 reactogenicity in terms of solicited AEs occurred in all trial patients (Fig. 2). Local events were normal to moderate (grade 0 to 2) in 87% of study patients. 94% of patients showed the expected formation of a granuloma/induration

at the injection site, which persisted beyond day 56. Severe AEs (grade 3) comprised local erythema in 11% of patients. 4% of patients reported localized inguinal lymphadenopathy. Local skin ulceration at the injection site was reported by 2% of patients. No fever or other systemic inflammatory solicited AEs were reported. Other systemic solicited AEs occurred in 26% of patients. 93% of the reported systemic solicited AEs were mild, with transient fatigue being the most frequently reported (17% of patients). One patient suffered from unrelated (chemotherapy-induced) grade 4 neutropenia. In 36% of patients, acute phase reaction with elevated C-reactive protein was observed until day 56 (Supplementary Note 1).

67 unsolicited AEs occurred, which were predominantly mild (79%, Supplementary Table 4, Supplementary Note 2). Of all unsolicited AEs, three were judged to be related to CoVac-1, comprising two viral reactivations (herpes simplex and varizella zoster virus) and one formation of a blister at injection site (grade 1). Of the unsolicited AEs, three were reported as SAE not related to CoVac-1 application (details in the Supplementary Note 2, Supplementary Table 4).

No immune-mediated AE was observed in any of the patients. Until day 56, two SARS-CoV-2 infections occurred, with reportedly mild disease course and resolved without sequel (Supplementary Note 3). Details on interim assessment of Phase I are provided in Supplementary Note 4.

### Immunogenicity

Immunogenicity was determined in terms of T-cell responses to the six SARS-CoV-2 HLA-DR CoVac-1 T-cell epitopes[9,10,17] using enzyme-linked immunospot (ELISPOT) assays. T-cell responses were assessed in all eligible patients at baseline (day 1), on day 7, day 14, and day 28 after CoVac-1 application. The immunogenicity endpoint was reached: CoVac-1-induced interferon (IFN)-γ T-cell responses were documented in 83% (95% CI 66–93%) of study patients within Phase II (93% (95% CI 66–100%) Phase I, 86% (95% CI 73–94%) Phase I/II combined) on day 28, with a 32-fold increase (median positive calculated spot counts: 5 (day 1) to 159 (day 28)) from baseline (Fig. 3a–c). Study patients unable to mount a CoVac-1-induced T-cell response possess significantly lesser HLA-DR allotype alleles matching those CoVac-1 peptides were

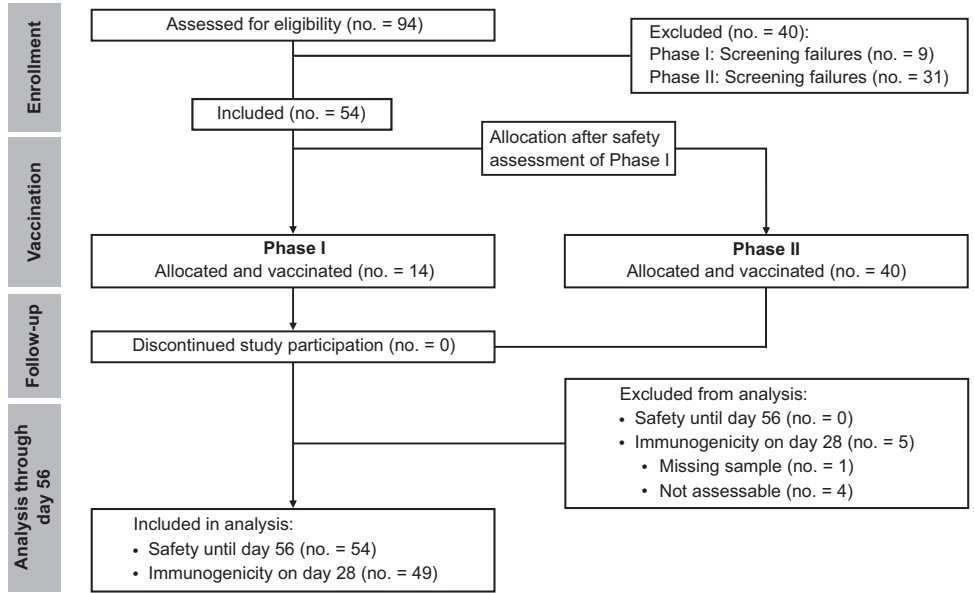

**Fig. 1 | Consort flow diagram of the trial.** 40 patients did not meet the inclusion criteria at screening and accordingly were not enrolled in the trial. All 54 enrolled patients received one dose of CoVac-1. Safety oversight to proceed to Phase II was performed by an independent data and safety monitoring board and approved by the competent authority (Paul Ehrlich Institute) and the local Ethics Committee after an interim safety and immunogenicity analysis of study patients included in Phase I, evaluated on day 28 after CoVac-1 application. Five patients were not assessable for the primary endpoint analysis of immunogenicity. All enrolled patients were assessable for safety. no., number.

**Table 1 | Patients' characteristics**

| Characteristics | All | Phase I | Phase II |
|---|---|---|---|
| Patients [no.] | 54 | 14 | 40 |
| Diagnosis [no. (%)] | | | |
| Primary immunodeficiency | | | |
| CVID | 2 (4) | 1 (7) | 1 (3) |
| XLA | 1 (2) | 1 (7) | 0 (0) |
| Others[a] | 1 (2) | 0 (0) | 1 (3) |
| Secondary immunodeficiency | | | |
| CLL | 16 (30) | 4 (29) | 12 (30) |
| MCL | 13 (24) | 3 (21) | 10 (25) |
| FL | 11 (20) | 4 (29) | 7 (18) |
| DLBCL | 5 (9) | 0 (0) | 5 (13) |
| Others[b] | 5 (9) | 1 (7) | 4 (10) |
| Age [years] | | | |
| Median | 61.8 | 62.5 | 61.5 |
| Range | 37–90 | 40–80 | 37–90 |
| Sex [no. (%)] | | | |
| Female | 15 (28) | 4 (29) | 11 (27) |
| Male | 39 (72) | 10 (71) | 29 (73) |

Assessment was done at the time of screening.

CLL chronic lymphocytic leukemia, CVID common variable immunodeficiency, DLBCL diffuse large B-cell lymphoma, FL follicular lymphoma, MCL mantle cell lymphoma, no. number, XLA X-linked agammaglobulinemia.

[a]Unspecified agammaglobulinemia.

[b]Hodgkin's lymphoma, marginal cell lymphoma, myeloproliferative syndrome, Waldenström's macroglobulinemia.

selected for (Fig. 3d). In silico binding predictions demonstrate that the CoVac-1 peptides could bind to multiple further HLA class II allotypes beside the HLA-DR allele mainly designed for, in detail CoVac-1 peptides are predicted to bind to 2717, 4096, and 10,540 different HLA-DR, -DP, or -DQ allotypes, respectively. Vaccine-induced T-cell responses were directed to multiple CoVac-1 peptides with median 4/6 peptides recognized by patients' T cells on day 28 (Fig. 3e) with no relevant association to the number of matching HLA-DR alleles. The CoVac-1 peptide P6_ORF8 most frequently induced T-cell responses (75%), followed by P3_spi (69%), P5_mem and P4_env (both 52%, Supplementary Fig. 1). 12% and 44% of the study patients showed low-frequent pre-existing SARS-CoV-2 T-cell responses ex vivo and after 12 days in vitro stimulation at baseline, respectively, in particular to the spike-derived peptide P3_spi (Supplementary Figs. 1 and 2). This may be explained by prior vaccination with approved COVID-19 vaccines as well as by cross-reactive T-cell responses to human common cold coronaviruses (HCoV-OC43, HCoV-229E, HCoV-NL63, HCoV-HKU1)[9,16].

CoVac-1-induced CD4$^+$ T cells displayed a multifunctional T-helper 1 (Th1) phenotype with positivity for IFN-γ, tumor necrosis factor (TNF), interleukin-2 (IL-2), and CD107a (Fig. 4a). No CD8$^+$ T-cell responses could be detected ex vivo. Frequency of functional CD4$^+$ T cells was increased up to 38-fold after in vitro expansion (e.g., 0.03% (ex vivo) to 1.15% (median positive samples) CoVac-1-specific CD107a$^+$CD4$^+$ T cells), indicative of potent expandability of the induced T cells upon SARS-CoV-2 exposure (Supplementary Fig. 3). In two patients a low frequent CoVac-1-specific IFN-γ$^+$ (0.07%) or TNF$^+$ (0.09%) CD8$^+$ T-cell response was detectable after in vitro expansion.

Subgroup analysis points towards higher response rates and frequencies of CoVac-1-induced T cells on day 28 for patients with acquired B-cell deficiency (87% response rate, median calculated spot count 151) compared to patients with congenital B-cell deficiency (75% response rate, median calculated spot count 81), with highest frequencies of T cells observed in patients with FL and diffuse large B-cell lymphoma (DLBCL) (median calculated spot counts 401 and 663, respectively, Fig. 4b). No relevant difference in the frequency and intensity of CoVac-1-induced T-cell responses was observed between cancer patients with or without ongoing anti-CD20 therapy (85% vs 88% response, median calculated spot count 151 vs. 164, Fig. 4b).

Beyond T-cell responses, a seroconversion in terms of induction of low-level SARS-CoV-2 anti-spike IgG antibodies was observed in single patients ($n = 8$) on day 28 (Fig. 4c). No differences in CoVac-1-induced T-cell response intensity was observed between patients with and without seroconversion (median calculated spot counts 75 (with seroconversion) to 86 (without seroconversion)).

The intensity of CoVac-1-induced IFN-γ T-cell responses (median calculated spot count 144) exceeded spike-specific T-cell responses induced by approved mRNA-based vaccines (median calculated spot count 45, median time after mRNA vaccination 66 days) in B-cell deficient patients prior to CoVac-1 application (Fig. 5a, Supplementary Table 7). Such pre-existing spike-specific T-cell responses after mRNA vaccination were boosted by CoVac-1, and expanded to various CoVac-1 peptides derived from other SARS-CoV-2 proteins (Fig. 5b, c). Notably, none of the variant-defining or associated mutations of the Omicron variants (BA.1, BA1.1, BA.2, BA.3, Fig. 5d, Supplementary Table 9)[19] affected any of the CoVac-1 peptides.

The intensity of CoVac-1-induced IFN-γ T-cell responses in B-cell deficient patients at day 28 (B-CoVs, median calculated spot count 144) was similar or even higher than T-cell responses to CoVac-1 peptides (median calculated spot count 55), to SARS-CoV-2-specific (median calculated spot count 61) and to cross-reactive (median calculated spot count 105) T-cell epitopes[9,10] in immunocompetent healthy COVID-19 convalescents (HCs), with asymptomatic and mild disease[9,17] (Fig. 5e, Supplementary Table 6). The same held true for the comparison of CoVac-1-induced T-cell responses in B-cell deficient patients with a cohort of immunocompetent HCs[9] with asymptomatic and mild disease that did not develop a humoral anti-spike IgG response upon infection (Fig. 5f), indicating that CoVac-1-induced T-cell responses in B-cell deficient patients might be sufficient to provide immunity against severe COVID-19 in this highly immunocompromised population.

## Discussion

Recently, evaluation of the peptide-based T-cell activator CoVac-1 in healthy adults showed promising safety and immunogenicity in terms of profound and long-lasting SARS-CoV-2-specific T-cell responses that are not affected by VOCs[17]. This prompted clinical evaluation in patients with congenital or acquired B-cell deficiency, the latter comprising patients receiving B-cell depleting therapy, e.g., in leukemia and lymphoma. This patient population is unable to mount sufficient humoral immune response upon vaccination with approved vaccines[6–8] and is at high risk for a severe course of COVID-19[20–24].

Here, we report on the Phase I/II clinical trial evaluating CoVac-1 in patients with congenital or acquired B-cell deficiency, which confirmed the favorable safety profile and documented potent de novo induction of T-cell responses after one single administration even in this highly immunocompromised study population. Of note, beyond B-cell deficiency, 50% of patients additionally presented with CD4$^+$ T cell counts below 500/µl, further emphasizing the severe immunodeficiency of the trial population[25]. Local granuloma formation was observed in 94% of study subjects displaying an expected and intended local reaction after Montanide administration[26,27], which enables continuous local stimulation of SARS-CoV-2-specific T cells required for induction of long-lasting T-cell responses without systemic inflammation[28]. In line with the findings in healthy volunteers[17], CoVac-1 induced a Th1 CD4$^+$ T-cell response in the B-cell deficient patients, precluding the theoretical risk of vaccine-associated enhanced respiratory disease, which has been associated with a T-helper 2 (Th2)-driven immune response[29].

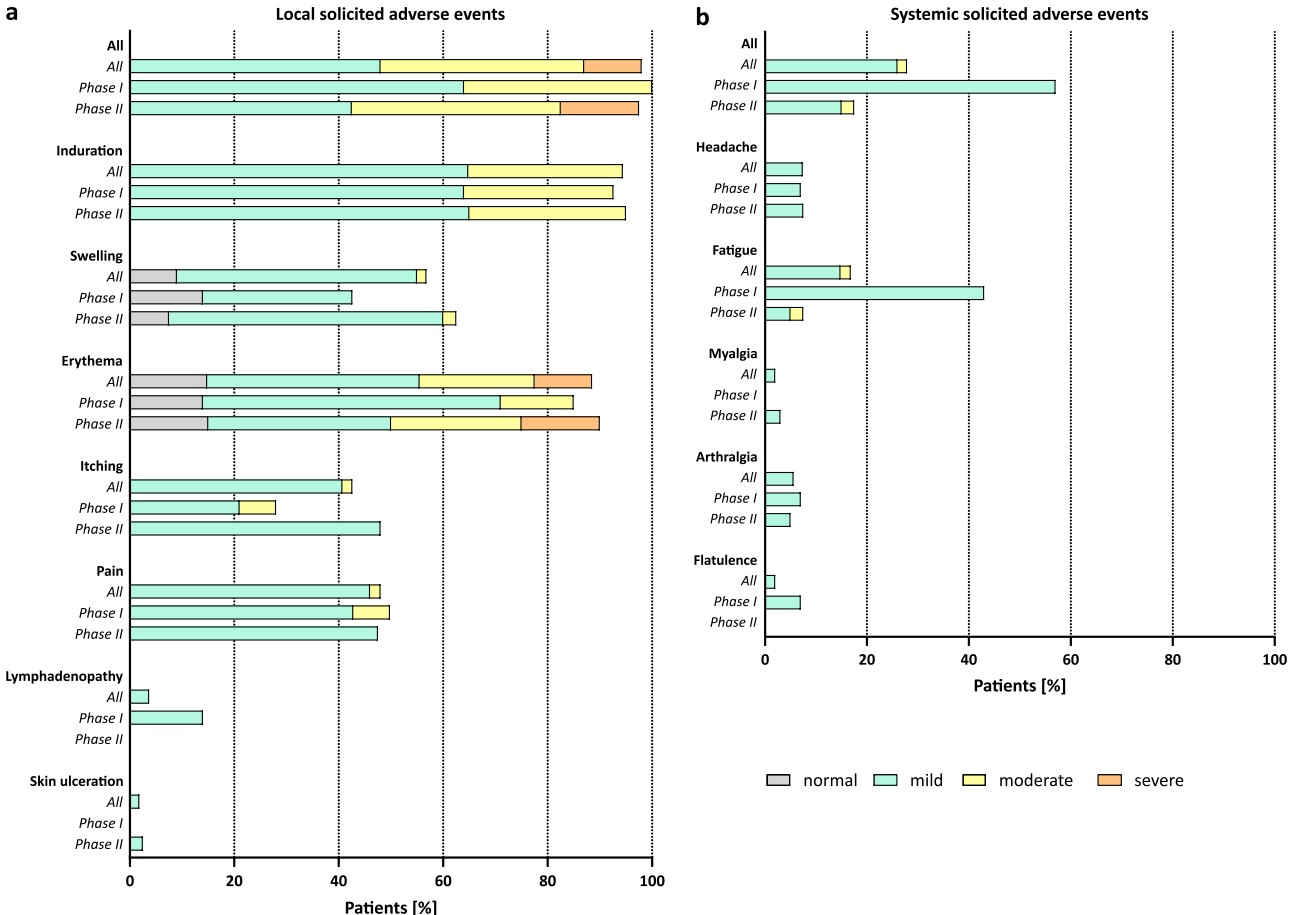

**Fig. 2 | Local and systemic solicited adverse events.** Related (**a**) local and (**b**) systemic solicited AEs documented within 56 days after CoVac-1 administration. Severity was graded as normal (grade 0), mild (grade 1), moderate (grade 2), or severe (grade 3) based on the definition provided in the methods section and the Supplementary Information.

The T-cell activator CoVac-1 was designed to primarily activate CD4[+] T-cell responses as previous data showed the relevance of CD4[+] T-cell response in SARS-CoV-2 convalescent individuals[9,30]. Nevertheless, the CoVac-1 HLA-DR T-cell epitopes also contain embedded HLA class I sequences for the additional induction of CD8[+] T-cell responses, although similar to a previous Phase I clinical trial in healthy individuals[17] this occurs at a considerably lower frequency. CoVac-1-induced T-cell responses showed high diversity targeting multiple vaccine peptides derived from different viral proteins, which is of high relevance for anti-viral defense and disease outcome in viral infections including SARS-CoV-2[9,18,31,32]. The broad T-cell responses induced by CoVac-1 are not affected by any of the current SARS-CoV-2 VOCs[17], including the latest Omicron variants[19], which are associated with loss of neutralizing antibody capacity and reduced efficacy of approved vaccines[33,34].

CoVac-1-induced T-cell responses exceeded spike-specific T-cell responses induced by mRNA vaccination[2,3,35] in B-cell deficient patients. Moreover, CoVac-1 was able to boost such pre-existing spike-specific T-cell responses.

Subgroup analysis points towards higher frequency and intensity of T-cell responses in cancer patients with acquired B-cell deficiency compared to patients with congenital B-cell deficiency, however, the small size of the primary immunodeficiency cohort limits this conclusion. This could be the consequence of additional T-cell defects in patients with congenital immune defects (e.g., Common Variable Immunodeficiency, CVID)[36,37].

Even after receipt of two or more doses of approved vaccines, none of the patients showed any humoral immune response to SARS-CoV-2 at study inclusion; after CoVac-1 application, induction of low-level SARS-CoV-2 anti-spike IgG antibodies was observed in single patients, despite consistently negative results in sequential SARS-CoV-2 PCRs. This finding may be due to stimulation of pre-existing[38] or vaccine-induced low frequent B cells by CoVac-1-induced CD4[+] T cells.

T cell-mediated immunity and, in particular CD4[+] T cells, are indispensable for the generation of protective antibody responses, reinforcement of CD8[+] T-cell responses[39,40], as well as direct killing of virus-infected cells[41,42]. Repetitive application of approved COVID-19 vaccines has been shown to induce a spike-specific T-cell response even in patients with B-cell deficiency[43,44]. The relevance of anti-viral T-cell responses during acute infection and for long-term immunity was also proven specifically for SARS-CoV-2[9,10,13–16]. Moreover, cases of asymptomatic SARS-CoV-2 infection, as well as reports from patients with congenital B-cell deficiency document cellular immune responses without seroconversion, providing evidence for the role of T-cell immunity in disease control, even in the absence of neutralizing antibodies[14,45]. Accordingly, CoVac-1 may well serve as a T-cell activator beyond current prophylactic approaches in immunocompromised patients comprising substitution of immunoglobulins and application of monoclonal SARS-CoV-2 antibody products[46,47].

To elucidate which frequencies and phenotypes of T cells are required to effectively combat COVID-19 and to what extend CoVac-1-induced T-cell responses are protective for severe disease, a longitudinal Phase III efficacy study with CoVac-1 is presently in preparation. Evidence that CoVac-1-induced T-cell responses may confer immunity to severe COVID-19 in this highly immunocompromised population is provided by the phenotype of CoVac-1-induced T cells,

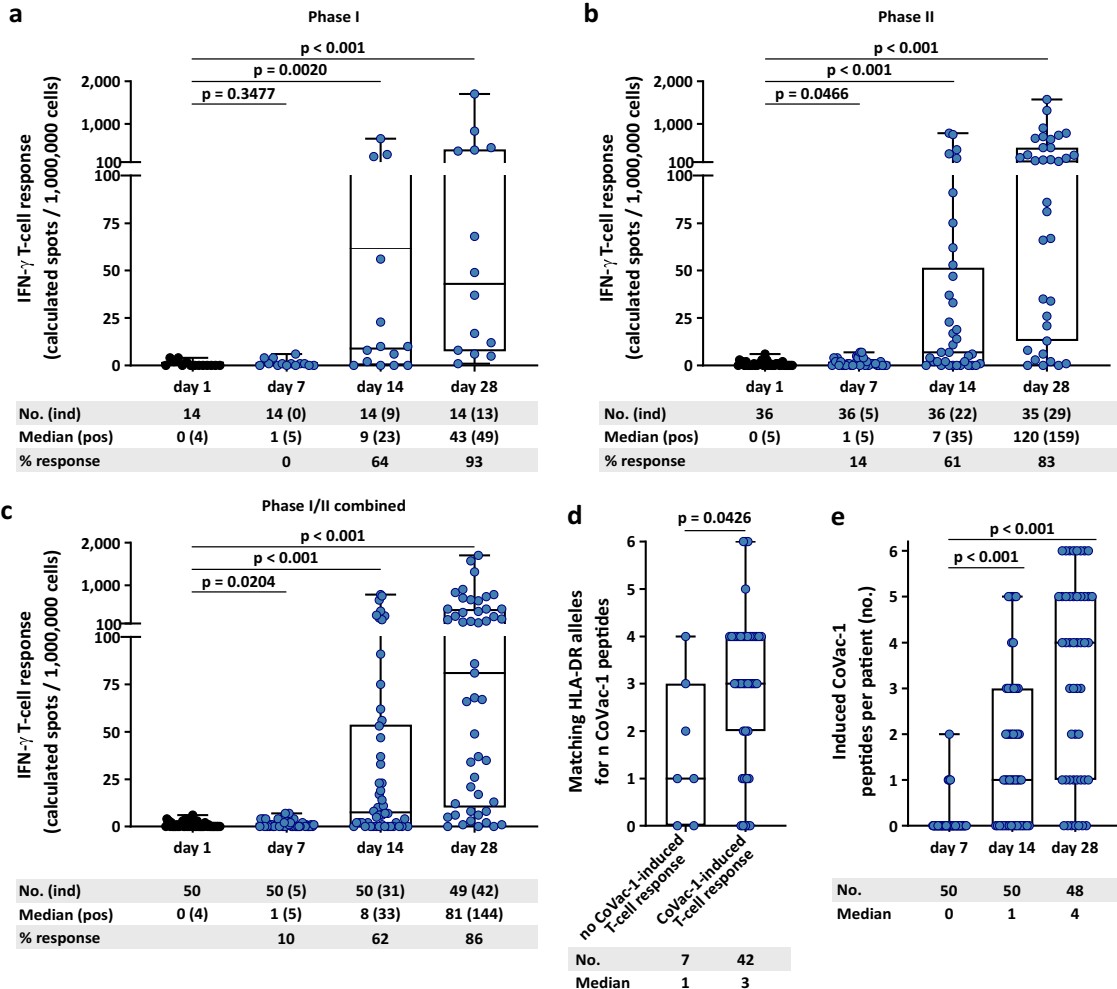

**Fig. 3 | CoVac-1-induced T-cell responses.** CoVac-1-induced T-cell responses assessed ex vivo by IFN-γ ELISPOT assays using PBMCs from (**a**) Phase I, (**b**) Phase II, (**c**) Phase I/II combined study patients (*n* = as indicated) collected before administration (day 1) and at different time points after administration (day 7, day 14, day 28). No. (ind) indicates the number of analyzed patients with the number of patients showing a CoVac-1-induced T-cell response (spot count post vaccination ≥2-fold higher than the respective spot count on day 1) in brackets. Median (pos) represents the median intensity of T-cell responses with the median of positive T-cell responses in brackets (≥3-fold higher than the negative control). % response indicates the percentage of study patients showing CoVac-1-induced T-cell responses. Two-sided Wilcoxon signed-rank test. **d** Number of CoVac-1 peptides for which patients without and with CoVac-1-induced T-cell responses at day 28 after CoVac-1 application possess matching HLA-DR alleles. Two-sided Mann-Whitney U test. **e** The number of CoVac-1 T-cell epitopes (*n* = 6) per patient (*n* = as indicated) that elicited a vaccine-induced T-cell response. Two-sided Wilcoxon signed-rank test. (**a**, **d**) The intensity of T-cell responses is depicted as calculated spot counts (mean spot count of technical replicates minus the respective negative control). (**a**–**e**) Box plots show median with 25th or 75th percentiles, and min/max whiskers. no., number; pos, positive; ind, induced.

which resembles that upon natural infection[9–11], and the intensity of the CoVac-1-induced T-cell responses, which is similar and even higher compared to a cohort of immunocompetent HCs with asymptomatic and mild COVID-19 that did not develop a humoral anti-spike IgG response upon infection and thus likely were protected by a SARS-CoV-2-specific T-cell response alone. Despite lacking a humoral immune response to SARS-CoV-2, two study patients experienced a mild course of COVID-19 after CoVac-1 application.

Limitations of our trial include the small sample size, low ethnic diversity and exclusion of patients with autoimmune disease receiving B-cell-depleting therapies. Safety and immunogenicity data over a longer observation period are presently being collected during the study follow-up where patients are monitored for up to 6 months.

In conclusion, the safety and immunogenicity data of our trial demonstrate that CoVac-1 is a promising T-cell activator for induction of SARS-CoV-2 T-cell immunity in patients with congenital or acquired B-cell defects and warrant advancement to a pivotal Phase III safety and efficacy evaluation.

## Methods

### Trial design and oversight

The multi-center Phase I/II trial (ClinicalTrials.gov Identifier: NCT04954469) was conducted at the Clinical Collaboration Unit (CCU) Translational Immunology, University Hospital Tübingen, the Institute of Clinical Cancer Research, Krankenhaus Nordwest, University Cancer Center, Frankfurt and the Department of Hematology, Oncology and Cancer Immunology, Campus Benjamin Franklin, Charité-Universitätsmedizin Berlin, Germany. Men as well as non-pregnant women aged ≥18 years, with congenital or acquired B-cell deficiency, defined by either decreased IgG serum concentration, ongoing immunoglobulin substitution for hypogammaglobulinemia, or ongoing or following (up to six months) treatment regimens containing anti-CD20 immunotherapy (e.g., rituximab) were eligible. Patients were enrolled independently of COVID-19 vaccination status if negative for anti-spike SARS-CoV-2 antibodies at the time of inclusion. Individuals with history of SARS-CoV-2 infection (real-time polymerase chain reaction (PCR) or antibody test)

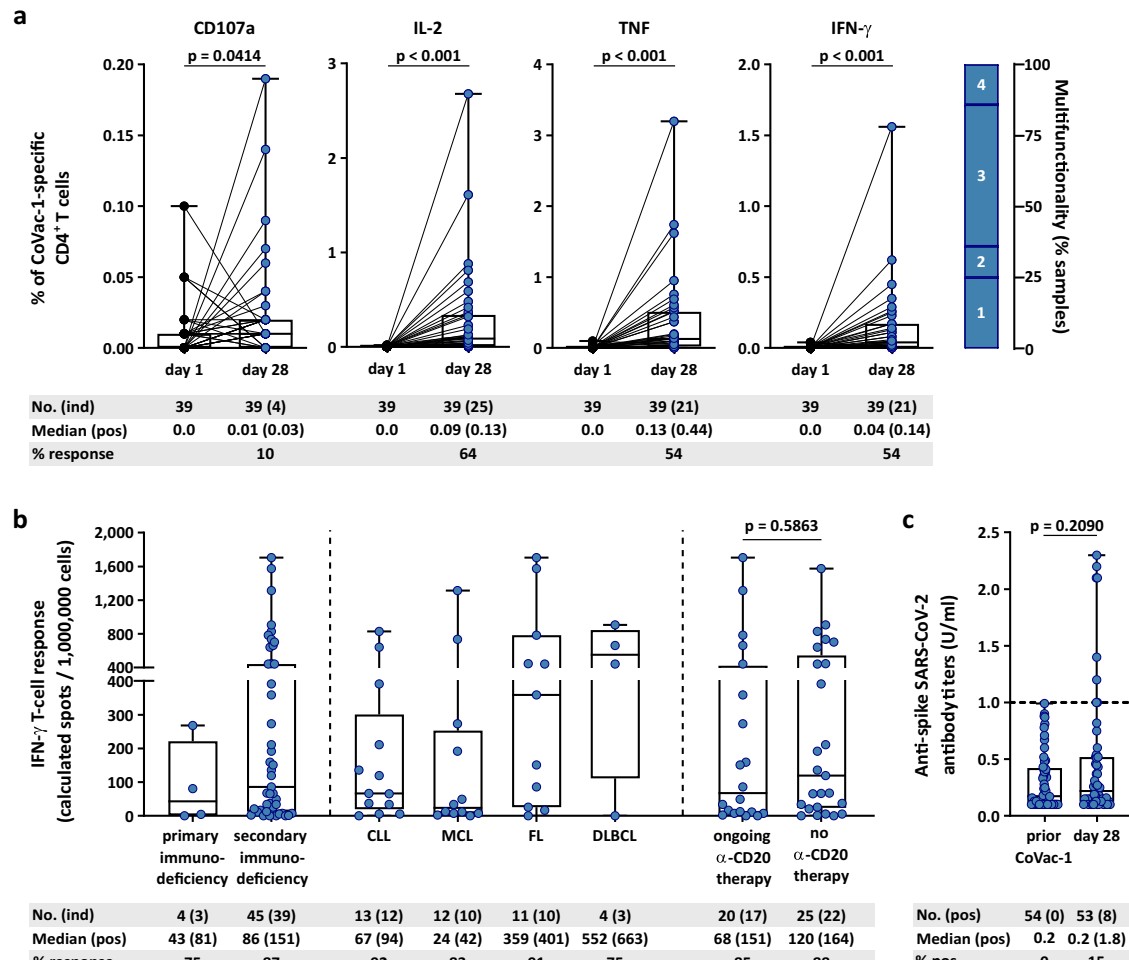

**Fig. 4 | Characterization of CoVac-1-induced T-cell responses. a** Frequencies of functional CoVac-1-induced CD4+ T cells in study patients (*n* = as indicated) prior to administration (day 1) and at day 28 after CoVac-1 application using ex vivo intracellular cytokine (IFN-γ, TNF, IL-2) and cell surface marker staining (CD107a). The right graph displays the proportion of samples revealing monofunctional (1), difunctional (2), trifunctional (3), or tetrafunctional (4) CD4+ T cells. Two-sided Wilcoxon signed-rank test. **b** Subgroup analysis (*n* = as indicated) of IFN-γ T-cell responses assessed on day 28 according to the type of immunodeficiency (primary vs. secondary), the type of hematologic malignant diseases (CLL, MCL, FL, DLBCL) and treatment (ongoing vs. no (without or discontinued (>six months)) anti-CD20 antibody treatment). Two-sided Mann–Whitney *U* test. **c** Anti-spike IgG antibody titers assessed prior to administration and on day 28 after administration (*n* = as indicated). Values <0.1 were set to 0.1 and values ≥1.0 were considered positive. Two-sided Mann–Whitney *U* test. (**a–c**) Box plots or combined box-line plots show median with 25th or 75th percentiles, and min/max whiskers. CLL, chronic lymphocytic leukemia; DLBCL, diffuse large B-cell lymphoma; FL, follicular lymphoma; MCL, mantle cell lymphoma; no., number; pos, positive; ind, induced.

were excluded. A detailed description of inclusion and exclusion criteria is provided in the Supplementary Information. Health status was based on medical history and laboratory values, vital signs and physical examination at screening. With regards to sex and gender of patients, only sex was considered for this trial, which was based on self-reported assessment. Prior to enrollment, all patients provided written informed consent. Only eligible patients were recorded in the electronic case report form (eCRF). As safety and efficacy measure, a run-in Phase I trial was conducted with a follow-up period of 28 days after administration, followed by vaccination of further study patients in Phase II. Assessment of the run-in Phase I included the decision to proceed with a single CoVac-1 administration (detailed description in Supplementary Information). The trial was open-label without a control arm and funded by the Federal Agency of Science, Research and the Arts (BMBF), Germany and the Ministry of Science, Research and the Arts Baden-Württemberg (MWK), Germany. The trial was approved by the local ethics committees under the lead of the Ethics Committee at the University Hospital Tübingen (255/2021AMG1) and the competent authority Paul Ehrlich Institute and performed in accordance

with the International Council for Harmonization Good Clinical Practice guidelines.

Safety and immunogenicity assessment to proceed to Phase II was performed by an independent data safety monitoring board (DSMB).

### T-cell activator peptides and adjuvant
CoVac-1, developed and produced by the Good Manufacturing Practices (GMP) Peptide Laboratory at the Department of Immunology, University of Tübingen, Germany, is a peptide-based T-cell activator comprising six HLA-DR-restricted SARS-CoV-2 peptides[17] derived from various SARS-CoV-2 proteins (spike, nucleocapsid, membrane, envelope, and ORF8) and the synthetic lipopeptide adjuvant XS15, a TLR1/2 ligand[28,48] (manufactured by Bachem AG, Bubendorf, Switzerland) emulsified in Montanide™ ISA51 VG[26] (manufactured by Seppic, Paris, France).

CoVac-1 peptides (250 μg/peptide) and XS15 (50 μg) were prepared as water-oil emulsion 1:1 with Montanide™ ISA51 VG with an injectable volume of 500 μL. Each patient received one subcutaneous injection of CoVac-1 at the abdomen on day 1. The dose was based on the safety and efficacy data of a Phase I trial in healthy adults[17].

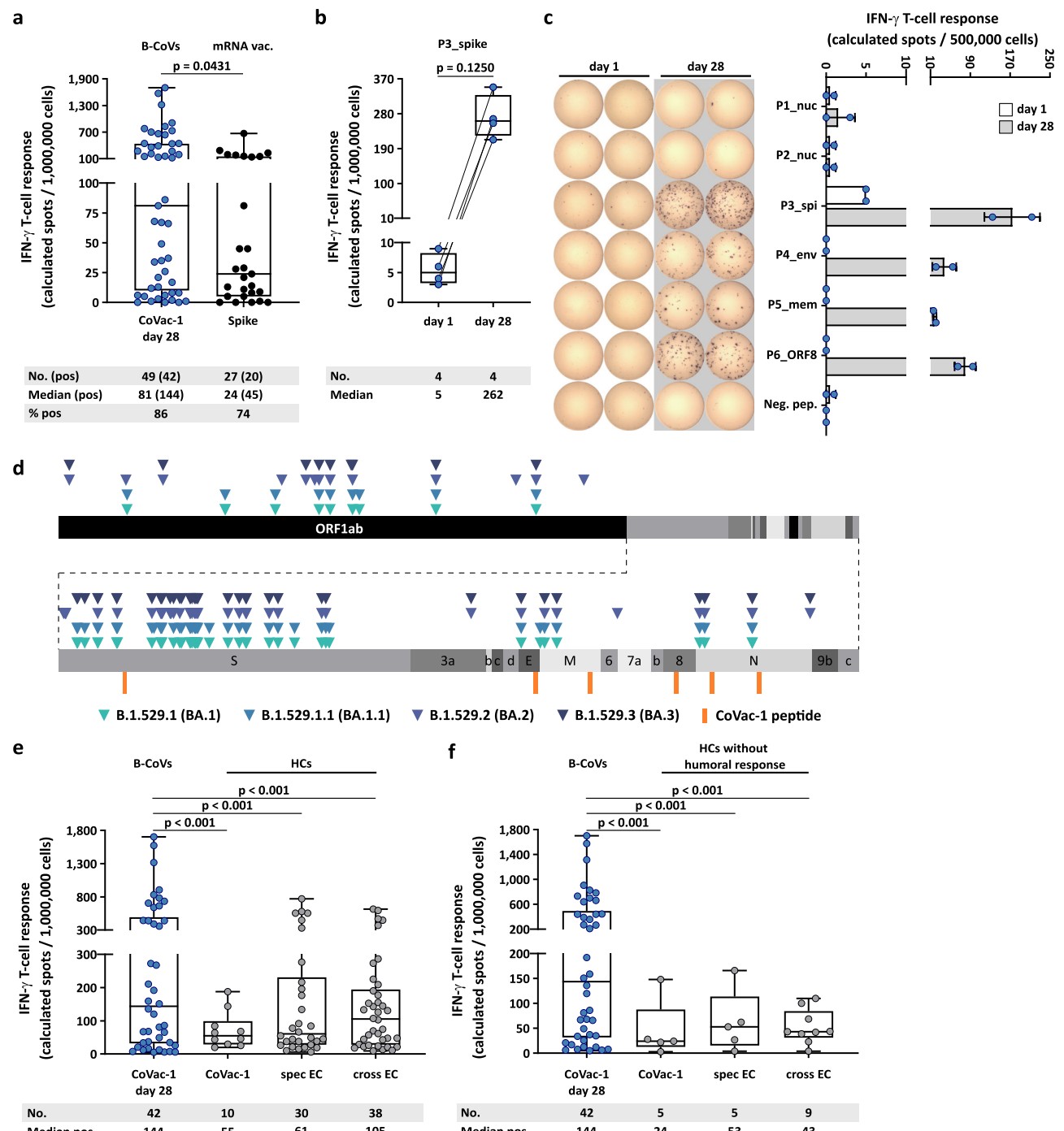

**Fig. 5 | CoVac-1-induced T-cell responses with regard to Omicron variants and compared to mRNA vaccine- or infection-induced T-cell response. a** CoVac-1-specific T-cell responses assessed ex vivo in study patients (day 28) compared to spike-specific T-cell responses prior to CoVac-1 administration in patients after second or third vaccination with approved mRNA vaccines (median time after mRNA vaccination 66 days, *n* = as indicated). Two-sided Mann–Whitney *U* test. **b** Intensities of P3_spike-induced IFN-γ T-cell responses assessed ex vivo in study patients that showed pre-existing P3_spike-specific T-cell responses (*n* = as indicated) prior to and on day 28 after CoVac-1 administration. Two-sided Wilcoxon signed-rank test. **c** Exemplary ex vivo ELISPOT assays of one study patient (UPN12), with pre-existing T-cell responses to P3_spike, for the six CoVac-1 peptides on day 1 (white) and day 28 (gray). The intensities of IFN-γ T-cell responses are depicted as calculated spot counts (mean spot count of technical replicates minus the respective negative control). **d** Color-coded mutations described for SARS-CoV-2 Omicron variants are shown together with CoVac-1 peptides (orange). Positive T-cell responses to specific (spec) and cross-reactive (cross) T-cell epitope compositions (ECs) in (**e**) immunocompetent HCs (CoVac-1, spec EC, cross EC, *n* = as indicated)[9,10] and (**f**) immunocompetent HCs without anti-SARS-CoV-2-antibody response after infection (CoVac-1, spec EC cross EC, *n* = as indicated) compared to positive IFN-γ T-cell responses in study patients assessed ex vivo (B-CoVs, *n* = as indicated, day 28). Two-sided Mann–Whitney *U* test. (**a**, **b**, **e**, **f**) The intensity of IFN-γ T-cell responses is depicted as calculated spot counts (mean spot count of technical replicates minus the respective negative control). Box plots or combined box-line plots show median with 25th or 75th percentiles, and min/max whiskers. **c** Bars with mean, SD and single data points. no., number; EC, epitope composition; HCs, healthy COVID-19 convalescents; ORF, open reading frame; pos, positive.

## Safety assessment

Primary safety outcomes reflect the nature, frequency, and severity of solicited AEs until day 56 after CoVac-1 application. Documentation was facilitated by a patient diary (covering 28 days after application) and graded by the investigators according to a modified Common Terminology Criteria for Adverse events (CTCAE) V5.0 grading scale (Supplementary Table 1). In addition, the number and percentage of trial patients with unsolicited events until day 56 was reported (according to CTCAE V5.0). Safety assessment included clinically significant changes in laboratory values (hematology and blood chemistry), SAE, and adverse events of special interest (AESI), which included SARS-CoV-2 infection, COVID-19 manifestations, and immune-mediated medical conditions (Supplementary Tables 1, 2 and 3).

## Immunogenicity assessment

The primary immunogenicity endpoint was defined by the induction of CoVac-1-specific T-cell responses in at least 70% of study patients to one or more of the CoVac-1 peptides or the combination of all six peptides, evaluated on day 7, day 14, and day 28 by IFN-γ ELISPOT assay ex vivo and after in vitro T-cell expansion. For T-cell expansion, peripheral blood mononuclear cells (PBMCs) were pulsed with CoVac-1 peptides (5 μg/mL per peptide) and cultured for 12 days adding 20 U/mL interleukin-2 (IL-2, Novartis) on days 3, 5, and 7. For IFN-γ ELISPOT (ex vivo or after in vitro expansion), cells were stimulated with 2.5 μg/mL of CoVac-1 peptides and analyzed in technical replicates. T-cell responses were considered positive (indicated as median of positive samples) if mean spot count was ≥ three-fold higher than mean spot count of negative control and defined as CoVac-1-induced (indicated as response (%)) if the mean spot count post administration was ≥ two-fold higher than the respective spot count on day 1 (baseline, prior to vaccination). Patients without the general ability to mount antigen-specific T-cell responses (absence of CoVac-1-induced T-cell responses and no T-cell responses to HLA-DR T-cell epitope control panel including Epstein-Barr virus (EBV), cytomegalovirus (CMV) and adenovirus (ADV) peptides as previously described[18], Supplementary Table 5) were considered as not assessable (drop-out). CoVac-1-induced T-cell responses were further characterized using cell surface markers and intracellular cytokine staining (ICS). For the latter, cells were stimulated with 10 μg/mL per peptide. The gating strategy is provided in Supplementary Fig. 4. Immunogenicity results were compared with those of immunocompetent HCs with PCR-confirmed SARS-CoV-2 infection (Supplementary Table 6). All assays were conducted in a blinded fashion and are described in detail in the Supplementary Information.

## Statistical analysis

The total sample size calculation (n = 54 patients) of the trial was based on the following assumptions: For the analysis of safety in the first 14 patients (Phase I), incidence of SAE associated with administration of CoVac-1 exceeding a predetermined rate of 20% was investigated. The trial was expanded to Phase II after proving safety and sufficient T-cell response (>80% of patients, with documented CoVac-1-induced T-cell responses) measured by IFN-γ ELISPOT on day 28. Here, the sample size (n = 40) based on the assumption that, in the unfavorable case of SARS-CoV-2-specific immune response induction in ≤50% of the patients, the treatment concept is extended with a probability of at most 5%. On the other hand, in the favorable case of peptide-specific immune response induction in ≥70% of patients, the concept would be followed with a probability of at least 80%. Safety data are summarized by counting every respective AE (lowest level term) that occurred in a patient only once. If the same AE occurred more than once, only the highest graded AE was counted. Data are displayed as mean ± standard deviation (SD), box plots as median with 25% or 75% quantiles and min/max whiskers. Details regarding statistical analysis plan and sample size calculation are provided in the Supplementary Information and the protocol.

## Reporting summary

Further information on research design is available in the Nature Portfolio Reporting Summary linked to this article.

## Data availability

Data supporting the findings of this study including de-identified patient data are available after final completion of the trial report and are shared according to data sharing guidelines upon reasonable request to the corresponding author J.S.W (juliane.walz@med.uni-tuebingen.de). Data will be only shared for non-commercial interests and after ethical approval. A data use agreement is obligatory.

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

## Acknowledgements

We thank all the patients of this trial and the members of the data and safety monitoring board (Peter Brossart, Bonn; Hansjörg Schild, Mainz; Clemens Wendtner, Munich). We are grateful to the technical and clinical staff of the participating trial centers at the CCU Translational Immunol-ogy, Department of Internal Medicine, University Hospital Tübingen, the Institute of Clinical Cancer Research (IKF), Krankenhaus Nordwest, Uni-versity Cancer Center, Frankfurt and the Department of Hematology, Oncology and Cancer Immunology, Campus Benjamin Franklin, Charité-Universitätsmedizin Berlin, the team of the "Wirkstoffpeptid Labor", Department of Immunology, Tübingen, the pharmacy of the University Hospital Tübingen, the data management team at the Institute for Clinical Epidemiology and applied Biometry, and the "Zentrum für Klinische Stu-dien", at the University Hospital Tübingen for support and coordination. We further thank EMC microcollections GmbH for provision of XS15. This work was supported by the Ministry of Science, Research and the Arts Baden-Württemberg, Germany (MWK, Sonderfördermassnahme COVID-19, TÜ17 J.S.W.), Federal Agency of Science, Research and the Arts, Ger-many (BMBF, FKZ:01KI20130 and FKZ:16LW0004K, J.S.W.), the Deutsche Forschungsgemeinschaft (DFG, German Research Foundation, Grant WA 4608/1-2, J.S.W.), the Deutsche Forschungsgemeinschaft under Ger-many's Excellence Strategy (Grant EXC2180-390900677, H.-G.R., H.R.S., and J.S.W.), the German Cancer Consortium (DKTK, H.-G.R., H.R.S., and J.S.W.), the Wilhelm Sander Stiftung (Grant 2016.177.2, J.S.W.), the José Carreras Leukämie-Stiftung (Grant DJCLS 05 R/2017, J.S.W.), the Robert Bosch Stiftung (C.M., S.U.J.), the BIH-Charité Clinician Scientist Program funded by the Charité –Universitätsmedizin Berlin and the Berlin Institute of Health (S.M.R., S.H., and C.A.P.), the Ernst-Jung-Preis für Medizin and the Landesforschungspreis Baden-Württemberg both awarded to H.-G. Rammensee. We acknowledge support from the Open Access Publication Fund of the University of Tübingen.

## Author contributions

J.S.H., K.-H.W., H.-G.R., H.R.S and J.S.W. were involved in the design of the overall study and strategy. C.M., I.F. and M.W.L. provided feedback on the study design. C.T., A.N., Y.M., J.B., J.R., M.W., S.M.S., N.H.G., M.M., A.H., B.K., K.L.C., M.L. and S.H. performed the immunogenicity analyses. J.S.H., M.M., T.H., S.M.R., C.M.T., S.U.J., S.H., A.P., E.J. and J.S.W.

conducted patient data and sample collection as well as medical evaluation and analysis. J.S.H., M.M., T.H., S.M.R., C.M.T., S.U.J., C.A.P., S.H., T.O.G., E.J., H.R.S. and J.S.W. collected data as study investigators. J.S. performed the in silico peptide binding prediction. S.U.J., C.M. and I.F. developed the statistical design and oversaw the data analysis. M.D., M.R. and M.T.O. conducted GMP production of CoVac-1. J.S.H., C.T., M.M., A.N., H.R.S. and J.S.W. drafted the manuscript. All authors supported the creation of the manuscript by review and editing.

## Funding

## Competing interests

J.S.H., H.R.S., A.N., H.-G.R. and J.S.W. are listed as inventors on a patent related to the SARS-CoV-2 T-cell epitopes included in CoVac-1. H.-G.R. and K.-H.W. are listed as inventor on a patent related to the adjuvant XS15 included in CoVac-1. The other authors declare no competing interests.

## Additional information

Jonas S. Heitmann [1,2,20], Claudia Tandler[2,3,4,20], Maddalena Marconato[1,2,20], Annika Nelde [2,3,4,20], Timorshah Habibzada[5], Susanne M. Rittig[6,7], Christian M. Tegeler[1,8], Yacine Maringer [2,3,4], Simon U. Jaeger[1,9,10], Monika Denk[2,3,4,11], Marion Richter[2,3,4,11], Melek T. Oezbek[2,3,4], Karl-Heinz Wiesmüller[12], Jens Bauer [2,3,4], Jonas Rieth[2,3,4], Marcel Wacker [2,3,4], Sarah M. Schroeder [3,4,13], Naomi Hoenisch Gravel [2,3,4], Jonas Scheid [2,3,4,14], Melanie Märklin [1,2], Annika Henrich[1,2], Boris Klimovich[1,2], Kim L. Clar[1,2], Martina Lutz[1,2], Samuel Holzmayer[1,2], Sebastian Hörber [15], Andreas Peter[15], Christoph Meisner[16], Imma Fischer[17], Markus W. Löffler [2,4,10,11,18], Caroline Anna Peuker[6,7], Stefan Habringer [6,7], Thorsten O. Goetze[5], Elke Jäger[19], Hans-Georg Rammensee[2,4,11], Helmut R. Salih [1,2,21] & Juliane S. Walz [1,2,3,4,21] ✉

[1]Clinical Collaboration Unit Translational Immunology, German Cancer Consortium (DKTK), Department of Internal Medicine, University Hospital Tübingen, Tübingen, Germany. [2]Cluster of Excellence iFIT (EXC2180) "Image-Guided and Functionally Instructed Tumor Therapies", University of Tübingen, Tübingen, Germany. [3]Department of Peptide-based Immunotherapy, University and University Hospital Tübingen, Tübingen, Germany. [4]Institute for Cell Biology, Department of Immunology, University of Tübingen, Tübingen, Germany. [5]Institute of Clinical Cancer Research, Krankenhaus Nordwest, UCT-University Cancer Center, Frankfurt, Germany. [6]Department of Hematology, Oncology and Cancer Immunology, Campus Benjamin Franklin, Charité -Universitätsmedizin Berlin, Berlin, Germany. [7]Berlin Institute of Health at Charité - Universitätsmedizin Berlin, BIH Biomedical Innovation Academy, BIH Charité (Junior) (Digital) Clinician Scientist Program, Berlin, Germany. [8]Department of Obstetrics and Gynecology, University Hospital Tübingen, Tübingen, Germany. [9]Dr. Margarete Fischer-Bosch Institute for Clinical Pharmacology, Stuttgart, Germany. [10]Department of Clinical Pharmacology, University Hospital Tübingen, Tübingen, Germany. [11]German Cancer Consortium (DKTK) and German Cancer Research Center (DKFZ), partner site Tübingen, Tübingen, Germany. [12]EMC microcollections GmbH, Tübingen, Germany. [13]Department of Otorhinolaryngology, Head & Neck Surgery, University Hospital Tübingen, Tübingen, Germany. [14]Quantitative Biology Center (QBiC), University of Tübingen, Tübingen, Germany. [15]Institute for Clinical Chemistry and Pathobiochemistry, Department for Diagnostic Laboratory Medicine, University Hospital Tübingen, Tübingen, Germany. [16]Robert Bosch Hospital, Robert Bosch Society for Medical Research, Stuttgart, Germany. [17]Institute for Clinical Epidemiology and Applied Biometry, University Hospital Tübingen, Tübingen, Germany. [18]Department of General, Visceral and Transplant Surgery, University Hospital Tübingen, Tübingen, Germany. [19]Department for Oncology and Hematology, Krankenhaus Nordwest, UCT-University Cancer Center, Frankfurt, Germany. [20]These authors contributed equally: Jonas S. Heitmann, Claudia Tandler, Maddalena Marconato, Annika Nelde. [21]These authors jointly supervised this work: Helmut R. Salih, Juliane S. Walz. ✉e-mail: juliane.walz@med.uni-tuebingen.de

