## [Peer review file · Nature Communications]

REVIEWERS' COMMENTS

Reviewer #1 (Remarks to the Author):

The authors have responded appropriately to the queries previously raised by the reviewers.

Reviewer #3 (Remarks to the Author):

Dr. Heitmann's team presented a phase I/II multicenter safety and immunogenicity trial of a multi-peptide vaccination to prevent COVID-19 infection in adults with B-cell/antibody deficiency. The concerns I raised about the absence of statistical tests and the incorrect reporting of the primary endpoint have been adequately addressed by Dr. Heitmann's team. Therefore, I have no further comments to add at this time.

Point-by-point reply:

Reviewer #1:

The authors have responded appropriately to the queries previously raised by the reviewers.

Author Reply: We thank you very much for your kind review and highly appreciate your positive evaluation of our revised manuscript.

Reviewer #2:

-

Reviewer #3:

Dr. Heitmann's team presented a phase I/II multicenter safety and immunogenicity trial of a multi-peptide vaccination to prevent COVID-19 infection in adults with B-cell/antibody deficiency. The concerns I raised about the absence of statistical tests and the incorrect reporting of the primary endpoint have been adequately addressed by Dr. Heitmann's team. Therefore, I have no further comments to add at this time.

Author Reply: We thank you very much for your kind review and highly appreciate your positive evaluation of our revised manuscript.